# Recent Advances in Hydrogel-Mediated Nitric Oxide Delivery Systems Targeted for Wound Healing Applications

**DOI:** 10.3390/pharmaceutics14071377

**Published:** 2022-06-29

**Authors:** Gina Tavares, Patrícia Alves, Pedro Simões

**Affiliations:** University of Coimbra, Chemical Process Engineering and Forest Products Research Centre, Department of Chemical Engineering, Rua Sílvio Lima, 3030-790 Coimbra, Portugal; ginatavares@eq.uc.pt

**Keywords:** Nitric oxide, hydrogel, wound dressing, chronic wounds

## Abstract

Despite the noticeable evolution in wound treatment over the centuries, a functional material that promotes correct and swift wound healing is important, considering the relative weight of chronic wounds in healthcare. Difficult to heal in a fashionable time, chronic wounds are more prone to infections and complications thereof. Nitric oxide (NO) has been explored for wound healing applications due to its appealing properties, which in the wound healing context include vasodilation, angiogenesis promotion, cell proliferation, and antimicrobial activity. NO delivery is facilitated by molecules that release NO when prompted, whose stability is ensured using carriers. Hydrogels, popular materials for wound dressings, have been studied as scaffolds for NO storage and delivery, showing promising results such as enhanced wound healing, controlled and sustained NO release, and bactericidal properties. Systems reported so far regarding NO delivery by hydrogels are reviewed.

## 1. Introduction—How Wound Care Is Still Relevant Nowadays

Wounds are ruptured and therefore structurally and physiologically compromised skin, caused by either trauma or physiological conditions. Depending on the time required to heal, wounds are typically categorized into acute or chronic [1]. Acute wounds tend to heal relatively fast, while chronic wounds take longer to properly heal. The latter most often arise from complications of specific diseases, with ulcers being the most common type of long-term wounds. Unfortunately, reoccurrence is a major issue in disease-caused chronic wounds, and it can only be avoided by the cure or management of the underlying disease [2]. Due to their high propensity to reappear, chronic wounds burden the healthcare system, and most importantly, negatively impact the quality of life of patients [3,4].

Chronic wounds (i.e., venous, arterial, pressure, and diabetic ulcers) have distinct causes but some characteristics in common, namely, infection, prolonged inflammatory phase, and biofilm formation. Cell abnormalities are also observed in chronic wounds, such as decreased growth factor receptors and reduced mitogenic potential, which impair cells from reacting to environmental signals [5]. These long-term wounds are more susceptible to infection, which further delays wound healing [6], and if left untreated, can cause impaired mobility, limb amputation, and eventually lead to death [7]. A study on the incidence of healthcare-associated infections in European Union countries, including Iceland, Norway, and the United Kingdom, during the period 2016–2017, showed that these infections easily exceeded 8 million per year, and over 3 million patients attained such infections each year at acute care hospitals [8]. Antimicrobial-resistant infections are responsible for large numbers of deaths (ca., 30,000 Europeans in 2020), and have been overshadowed by the still active coronavirus disease pandemic [9]. For further information regarding chronic wounds, we recommend reviews on this subject [4,5,7].

Wound healing is an elaborate cascade-like process that has four complex and overlapping phases (i.e., hemostasis, inflammation, proliferation, and remodeling [10]) that begin immediately after injury and last until re-epithelialization of the skin is completed [6,7] (Figure 1). Hemostasis starts immediately after injury and lasts between a few minutes to an hour. During this stage, platelets arrive to the wound bed, adhere to the extracellular matrix (ECM), and secrete proteins that initiate fibrin production and deposition, thus creating a clot that interrupts the bleeding. Throughout the process, these platelets also produce growth factors that attract neutrophils, macrophages, and fibroblasts to the wound bed. The next phase, inflammation, lasts for ca. 3 days, during which inflammation mediators increase the permeability of blood vessels and facilitate the arrival of neutrophils to the site. Neutrophils digest pathogens, foreign material, damaged cells, and ECM through phagocytosis. Once monocytes arrive and differentiate to macrophages, these instigate the proliferation of fibroblasts, smooth-muscle cells, and endothelial cells, hence beginning the proliferation stage. Starting around 48 h after injury, this phase consists of fibroblast proliferation, collagen and other ECM components production and deposition, and angiogenesis. The last phase, remodeling, starts 2–3 weeks after injury and can take several months to be completed. The ECM produced during the proliferation stage is remodeled by enzymes produced by fibroblasts, and lastly, macrophages and fibroblasts depart the wound site, ceasing the inflammation and proliferation stages [1,7].

Any imbalance in the process leads to impaired healing and chronic wound development. For instance, bacteria propagation enhances the inflammatory response and deeply compromises angiogenesis, thus impacting the amount of oxygen and nutrients capable of reaching the wound bed [10]. The evolution to chronic wounds can be prevented though a simple wound care regiment. Wound care is essentially performed in two steps: debridement and wound coverage. Debridement is the cleaning of the wound through removal of tissue debris that would otherwise be fuel for microbial proliferation and allows the exposure of healthy tissue to facilitate its proliferation [4,11]. The next step is the physical protection of the wound through the application of wound dressings.

Wound dressings are materials designed to cover damaged skin and are primarily meant to promote the wound healing process by acting as a pathogen penetration barrier while keeping the wound site moist through the absorption of excess exudate [12,13,14]. Moisture retention contributes to a proper and swift wound closure process as it aids the migration of new skin cells [4,15]. Deficient or excess exudate absorption is detrimental for proper wound healing as it leads to a microbial-friendly environment or dry wound site, respectively. Exudate levels differ during the healing stages, with high levels for the first 48 h [16]; thus, the main guideline is that wound dressings should ideally only absorb excess exudate without compromising the healing process [4]. Comparatively to normal skin, water loss is higher in wounds, reaching up to 5000 g·m^−2^d^−1^, around 20-fold the water loss of normal skin when at 35 °C. It has been disclosed that wound healing benefits from water vapor transmission values of around 2000–2500 g·m^−2^d^−1^ [17].

Since traditional wound dressings implicate constant reapplication and complementary methods to keep the wound aseptic, the development of wound dressings with attributes relevant for wound healing has been encouraged. Most research focuses on infection prevention, but besides antimicrobial properties, wound dressings can also be complemented with drugs or other components that accelerate the healing process, such as growth factors [18], anti-inflammatory drugs, and cytokines [19]. Growth factors, which are at low levels in chronic wounds, contribute to wound healing through the performance of several functions, including chemoattraction of macrophages, fibroblasts, and other cells; angiogenesis; and proliferation of fibroblasts and endothelial cells [18]. Cytokines have a role in wound healing and are responsible for inducing the migration of immune system cells to the wound site. Studies on the delivery of growth factors and cytokines showed increased wound closure [19].

For the past few decades, efforts have been made to confer antimicrobial properties to wound dressing supports such as foams, sponges, hydrogels, and gauzes. The simplest route is to load antimicrobial agents (e.g., silver [4], antibiotics, quaternary ammonium, and metallic nanoparticles [3]) into porous materials, but the materials can display antibacterial activity on their own (e.g., chitosan has been proven to have antibacterial activity [20]). Performance limitations of textile wound dressings (e.g., cotton or wool) have also fueled the search for different materials. Although soft in texture [3], textile wound dressings are devoid of the flexibility required for wounds located in mobility-related body parts such as joints [21]. When applied to burns, textile wound dressings adhere to the wound site in an uncontrollable manner, making its removal painful as the superficial layer of the wound bed is stripped in the process [1,22]. The ultimate wound dressings should be flexible [13], have antibacterial or at least bacteriostatic activity [21], exhibit adequate exudate absorption and gas permeability [13], allow a pain-free removal for the greater comfort possible, and be biocompatible [22,23,24,25]. The biocompatibility of a material is ascertained after an extensive array of tests that study the physical, chemical, and mechanical properties of the material as well as the potential adverse effects (i.e., allergenic, mutagenic, and cytotoxic) that may occur from its use, being crucial that the material does not elicit substantial damages or toxic effects to the body [26]. Research has shown that effective wound dressings exhibit porosity between 50% and 60%; these high values allow the transfer of oxygen and nutrients to the wound bed cells in contact with the dressing [17,23]. Pore size is also important since small pores physically hinder bacteria from reaching the wound site [11]. The interest is set on novel materials that intrinsically have a considerable number of desired properties (e.g., inherent antimicrobial activity and biocompatibility) and can perform well in wound environments. NO is a promising component regarding the design of ideal wound dressings. Due to its diminutive dimensions, this radical easily penetrates porous materials to reach the wound bed and triggers death cell mechanisms once it reaches bacterial membranes [27]. However, as a bioactive agent, NO demands storage and delivery vehicles. Therefore, it is theorized that hydrogels as vectors for the delivery of NO can be designed to fit the requirements of an excellent wound dressing. Polymers can form an assortment of materials that can accommodate the requirements of ideal wound dressings (e.g., foams, sponges, fibers and hydrogels), and due to a matrix similar to the extracellular matrix, nanofibers and hydrogels are the most explored polymeric materials for wound healing purposes [28]. The differentiating factor of NO-releasing hydrogels for wound healing is the functionality of NO. Antibacterial wound dressings on the market have an antimicrobial agent whose functionality is limited to preventing infections. NO, however, is unique because it participates in multiple aspects with regard to the evolution of the healing process besides infection prevention.

## 2. Hydrogels

Hydrogels are highly hydrated cross-linked polymers arranged in a matrix-like fashion that allow significant water retention (over 90% of their dry weight) in their three-dimensional network [29]. The most common natural polymers used for hydrogel formulation include collagen, alginate, hyaluronic acid [20], gelatin [30], cellulose [31], and chitosan [32]. Since most natural polymers already display biocompatibility and biodegradability [26], their high bioavailability further consolidates the proportion of interest in biopolymers over the past few decades [6]. However, biomedical applications are not exclusively reliant on natural polymers since many synthetic polymers are well-established biocompatible polymers (e.g., Poly(ethylene glycol) (PEG) [33] and Pluronic F-127 [34]) and are widely used for biomedical applications. Unlike natural polymers, synthetic polymers allow a greater degree of control over their composition. Biopolymers, however, require purification, and homogeneity is sometimes difficult to achieve due to different sources.

The structure and properties of hydrogels make them promising materials for the design of transdermal or injectable drug delivery systems, wound dressings, and adhesives [30,35]. Hydrogels are materials of great interest for wound healing due to their flexibility, adhesion, stability, and biodegradability, in addition to the capability of maintaining the wounded site moist, which helps to accelerate the healing rate [20,21,31,36]. Their porous extracellular matrix-like structure is also an important aspect to consider as it can facilitate the absorption of exudate from the wound bed [21,37] (Figure 2). Hydrogels can be modified to improve desired properties. For instance, knowing that hydrogel-tissue adhesion is limited in extreme wet conditions (e.g., bleeding), authors developed hemostatic hydrogels with enhanced tissue adhesion by grafting molecules that mimic adhesive components found in nature, namely methacrylate and dopamine [38].

Due to the characteristics mentioned above, hydrogels offer the possibility to simultaneously perform two functions, namely as a drug (or any bioactive agent relevant to wound closure) delivery system, and as a wound dressing [21]. In addition, hydrogels can be implemented as film/membranes [39], as a powder [40] (particles that gel in contact with liquid), or even be formed in situ (injectable [10,41,42,43]), making this class of materials highly convenient. For instance, hydrogel-forming powders better adapt to irregular wounds, and injectable hydrogels are excellent candidates for wounds located in mobility-related places [40,44]. In addition, powdering hydrogels has been reported as a route to patch and/or recycle mechanically damaged hydrogels. Powdered self-healing hydrogels regained their initial mechanical properties upon hydration [45].

## 3. Nitric Oxide and Its Donors

### 3.1. The Tiniest Antimicrobial Agent

Nitric oxide (NO) is a known bactericidal agent that has been explored for wound healing. It is effective towards a large range of bacteria, as well as fungi, parasites, and viruses [46,47,48]. NO stimulates the activity of immune cells at low concentrations (10^−12^–10^−9^ M) and promotes inhibition and death of pathogens at higher concentrations (10^−6^–10^−3^ M) [49,50,51]. NO reacts with superoxide to form peroxynitrite (NO_3_^−^), a very reactive oxidant responsible for membrane disruption via lipid peroxidation, and inactivation of enzymes via protein oxidation and nitration [52], phenomena that ultimately lead to bacteria death (Figure 3) [46]. Moreover, it has been reported that the concentration of NO considered lethal to bacteria (ca., 200 ppm of gaseous NO) is tolerable and non-toxic to human fibroblasts, which further validates the use of NO in the context of wound healing [47]. A study determined that gaseous NO at pressures above 200 ppm decreased cell viability and immune cell proliferation in mouse lymphocytes [51]. Furthermore, NO is a much safer alternative to typical antibiotics, as the overuse of antibiotics can trigger the development of resistance mechanisms in bacteria. Multidrug-resistant bacteria strains, as indicated by the term, are resistant to a variety of antibiotics [6], rendering these ineffective in the fight against bacteria proliferation. Recent studies report that NO alone is effective against a wide range of bacteria without the creation of resistance, and most importantly, in a clinical set, the susceptibility of drug-resistant bacteria to antibiotics is enhanced when the latter are complemented with NO [53]. In general, antimicrobial agents tend to have higher cytotoxicity than desired while antibiotics require higher concentrations to be effective [16]. Hence, NO’s adequacy for efficient antibacterial activity is supported by its synergetic antibacterial activity when allied to antibiotics, and although its cytotoxicity needs to be extensively studied, NO has inherent lower cytotoxicity compared to typical antimicrobial agents since NO is endogenously present in cells. 

The efficacy of an antibacterial agent differs from the conditions in which it is tested, namely against biofilm-inserted or planktonic bacteria. Bacteria can generate an extracellular matrix in which the diffusion of antibiotics is hindered, which complicates the fight against infections. More than the presence of planktonic bacteria, biofilm formation actually dictates if an acute wound becomes chronic [16] since biofilms are predominant in at least 60% of chronic wounds [55]. Biofilms allow gene transfer between bacteria, which could lead to the dissemination of genes associated with antibiotic resistance. Studies have shown that due to its size, NO diffusion is not hampered, and, most importantly, can disperse these films by restoring biofilm-incorporated bacteria to its planktonic state. NO-dispersed bacteria exhibit higher susceptibility to antibiotics [56]. Bacteria are more vulnerable in the free form, and most antibiotics are conceived to perform activity on planktonic bacteria [57]. Additionally, in respect to planktonic bacteria, biofilms require 2–10-fold NO and 1000–10,000-fold antibiotics to be destroyed. The ability to scatter biofilms makes NO a striking alternative to be considered in infection treatment and prevention. Moreover, NO directly kills biofilm-incorporated bacteria when the biofilm is exposed to high concentrations of NO [58].

Antibacterial activity is assessed by a variety of methods, through in vitro (inhibition zone on agar plates or inoculated broth in contact with the material), in vivo (i.e., mice), and ex vivo (excised animal skin) tests [59]. Anti-biofilm activity, as the ability of a material to destroy biofilms, is studied by allowing contact between a biofilm and the material being tested, followed by biofilm biomass determination through a staining protocol (e.g., crystal violet [60]) and determination of the viability of biofilm-embedded bacteria after exposure to the biocide (e.g., colony forming units count [61]). Depending on the method, additional information can be gathered; for example, by using a layered biofilm support, wound healing conditions can be mimicked and used to determine biocide efficacy (e.g., tetrazolium reduction used to assess cell viability) as well as its penetrability [62].

The most common bacteria detected in infected wounds are *Pseudomonas aeruginosa*, *Staphylococcus aureus*, *Klebsiella pneumoniae*, *Enterococcus faecalis*, *Acinetobacter baumannii* [55], and *Escherichia coli* [59]. Often, wounds are polymicrobial, and therefore, single-species biofilms might be inadequate to study the impact of antibacterial agents against typical in vivo infections. Some authors have reported the study of antibacterial efficacy in pluri-bacterial systems designed to better represent the bacterial composition and/or biocide susceptibility observed in of chronic wounds (i.e., *P. aeruginosa* and *S. aureus* biofilms) [59]. Although its use in wound healing is majorly due to its antimicrobial activity, NO has other properties that align with and further improve wound closure, such as promotion of angiogenesis, vasodilation, and fibroblast proliferation, among others [63]. Previous studies have shown that NO causes erythema when delivered topically owing to its vasodilator property. These tend to disappear a few minutes after NO delivery is interrupted [64]. NO has proven to direct endothelial differentiation of embryonic stem cells without growth factors by down-regulating pluripotent genes and up-regulating the expression of endothelial genes [65]. NO also accelerates endothelium proliferation, as demonstrated in a study with arterial grafts. Although no significant difference was observed after 3 months, NO-releasing arterial grafts exhibited greater endothelium coverage than the non-NO-releasing graft after the first month [66]. For further information about NO’s role in cell proliferation, the interested reader is referred to reference [67].

### 3.2. Nitric Oxide Donors

Owing to its gaseous nature and subsequent difficulty to effectively be stored and administered, NO is exogenously delivered to tissues, most commonly through NO donors. NO donors are any molecule or complex capable of releasing NO (e.g., organic nitrates and nitrites, metal–NO complexes, N-diazeniumdiolates, and S-nitrosothiols). Several factors must be weighted in the selection of an ideal NO donor for biomedical applications, the most relevant being the release mechanism and rate, and toxicity of the by-products following NO donation. Different applications demand distinct profiles, and concerning NO release rate, short-burst and prolonged releases are valuable for localized immediate effects and long-lasting effects in which NO is supposed to be continuously delivered, respectively [48]. It has been reported that wound healing benefits from a mixed release profile, i.e., a spontaneous short release followed by a continuous release. Higher NO concentrations at an early stage are important to hasten inflammation, which occurs in the first few hours, whereas the sustained delivery of NO throughout the ensuing stages of wound healing is beneficial for endothelial differentiation, which is characteristic of the last stage of wound closure [68,69].

With the ability to donate two or one NO molecule(s) per each parent molecule, *N*-diazeniumdiolates and *S*-nitrosothiols (RSNO) are the most attractive NO donors for biomedical applications [49]. Any aminated or thiolated molecules, such as polymers and peptides, can be converted to RSNO or *N*-diazenium-based donors through nitrosation, which greatly expands the range of possibilities for NO donor materials.

*S*-nitrosothiols are a group of molecules in which a nitroso group is bonded to a sulfur atom (RS-N=O), and are formed by the reaction of thiols with NO derivatives (e.g., NO_2_, N_2_O_4_, N_2_O_3_, and NO_2_^−^) [70,71]. *S*-nitrosoglutathione (GSNO) and *S*-nitrosocysteine (CySNO) are some of the most studied *S*-nitrosothiols with low molecular weight (Table 1). *S*-nitrosothiols are intermediates in biological processes, and thus stable in physiological conditions, namely at 37 °C and pH 7.4. However, the majority of RSNOs are easily decomposed at room temperature to form disulfides and NO, which limits their use [72]. These molecules release NO upon decomposition induced by enzymatic catalysis, light, and metal ions (i.e., Cu^2+^, Fe^2+^, Hg^2+^ and Ag^+^). NO release is accompanied by the formation of a disulfide, which is formed by the reaction between two thiyl radicals (RS•) [73]. The molecular structure of *S*-nitrosothiols influences the decomposition rate. Primary and secondary RSNOs are less stable, and thus exhibit a higher NO release rate. Moreover, these molecules can transfer the NO moiety to thiols without releasing NO, in a reaction termed trans-nitrosation [74,75], which in turn translates into a decreased possibility of generating peroxynitrite [48,75]. Due to this property, RSNOs have been linked to greater antibacterial activity as the result of nitrosylation of thiolated proteins [58].

N-diazeniumdiolates are a class of compounds generally termed NONOates due to the functional group [N(O)NO]^−^. *N*-diazeniumdiolates are stable molecules obtained by the reaction of secondary amines, present in both simple molecules and polymers, with NO at high pressures. Less common and less stable are NONOates formed with primary amines or amides, which in turn rely on protonated amines in the vicinity to achieve greater stability through hydrogen bonding formation [76,77]. NONOates undergo protonation to decompose into the parent amine and two molecules of NO, making NO release a pH-dependent process. These molecules are stable in basic media but undergo protonation at low pH to release NO. In other words, NO release is constrained in alkali media and triggered in acidic environments. The rate at which NO is released is entirely dependent on the donor structure (Table 2), and therefore subject to alterations. Diazeniumdiolates can be chemically modified at the second oxygen of the functional group, further contributing to the lengthening of NO release as it requires prior removal of the protecting moiety [47,78,79]. The potential of NONOates to form carcinogenic nitrosamines (R_2_-N-N=O) is the main shortcoming of the use of these molecules in biomedical devices [47]. An approach to counteract the formation of these undesired molecules has been explored and lies on the chemical attachment of diazeniumdiolates to a larger molecule (i.e., polymer). Under these conditions, some authors also hypothesize that the half-life of NONOates is altered and possibly prolonged [80].

Generally, NO donors are subject to burst releases and short half-lives. As previously mentioned, environmental manipulation (e.g., pH, temperature, light, and enzymatic degradation) and chemical modification can be used to tune the NO release rate. However, the physical protection of NO donors from external stimuli by vesicles or matrices is an option that has been extensively explored, as well. Polymer-based and lipid-based vesicles such as dendrimers, micelles, and liposomes are among the most common carriers for NO donors meant to extend NO release [49]. Matrices such as hydrogels and fibers can also be used to store and protect NO donors [69,82,83]. For instance, inserting *S*-nitrosothiols in hydrogels improves the stability of these molecules and prolongs NO release [84,85]. Moreover, some studies suggest that polymers with poor water solubility better protect NO donors that decompose in solution by limiting the interaction between water and the NO donor [86].

Besides NO-extended release, polymer-mediated NO delivery has been reported for enhanced antimicrobial activity. For instance, a synthetic antimicrobial copolymer modified with NO donating groups showed synergistic biofilm dispersal and antibacterial activity against *P. aeruginosa* [87]. Inspired by peptides with antimicrobial activity, antimicrobial polymers have been created to mimic its structure and properties—ideally, with cationic and hydrophobic segments [88]. These amphiphilic molecules adhere to and accumulate in bacteria cell membranes, disrupting cell integrity and leading to death [89].

## 4. NO-Releasing Hydrogel-Based Systems

### 4.1. Physically Adsorbed NO Donors

NO-delivering materials require protective measures to prevent precocious NO release (due to unwanted NO donor decomposition) prior to application. Hydrogels, matrices of excellence for wound healing, besides being adequate to store and protect NO donors, offer the possibility to be devoid of moisture for storage purposes, and can regain their structure upon hydration, relevant behavior when NO donors with hydrolytic NO release are used.

The incorporation of bioactive compounds into hydrogels can be achieved through a variety of methods and can be grouped according to hydrogel–drug interactions, namely chemical modification and physical adsorption [29,47,51]. NO release systems formed by NO donors or precursors covalently and non-covalently bound to a hydrogel matrix are summarized in Table 3 and Table 4. Most NO-releasing hydrogels were designed and tested for wound healing purposes, but due to the many properties of NO, other applications (i.e., antibacterial activity [41,90,91,92], vasodilation [39,93,94], biomedical applications or tissue engineering [65,92,95], anticancer activity [90,96,97]) were also targeted. NO is used in anticancer therapeutics as a way to make tumorous cells more susceptible to chemotherapeutic drugs. Poorly vascularized tumor masses produce HIF-1a (hypoxia-inducible factor), triggering cell resistance to death mechanisms (i.e., DNA damage, autophagy, and apoptosis) incited by radio or chemotherapy. Due to its angiogenic properties, NO normalizes tumor vasculature, thus ensuring the delivery of systemically administered drugs [98,99]. Oddly, NO plays a role in both encouraging and confining the proliferation of cancerous cells. Angiogenesis and proliferation facilitate cancer metastasis, whereas DNA damage leads to apoptosis [99]. As for tissue engineering applications, NO is advantageous because it can inhibit platelet adhesion and aggregation [100] on implanted materials, thus preventing blockages and subsequent cardiovascular complications. In other words, NO is implemented for its antimicrobial, angiogenic, and vasodilation properties, all important for the wound healing process.

Physically adsorbed NO donors are mainly based on small molecular weight RSNOs, GSNO being the most explored so far (Table 3). Even though a predilection is observed for natural polymers (e.g., gelatin, chitosan, and alginate), mostly due to the inherent biocompatibility and high bioavailability, the most explored polymer is Pluronic F-127, an amphiphilic poly(propylene oxide) and poly(ethylene oxide) co-polymer that easily forms micelles in solution [27]. A hydrogel system containing a metal–NO donor was tested for wound healing and promising results were obtained [82]. The light-activated NO release system was complemented with a coating for leaching prevention, thus allowing NO diffusion instead of the NO donor itself. Nitrite and organic nitrate-containing hydrogels meant to release NO were tested with special focus on the antibacterial activity of the systems, although the NO release profile of the systems was not explored in depth [83,113]. NO donors based on nitrosamines were also reported. A hydrogel with BNN6 exhibited excellent properties, namely NO release, antibacterial activity, mechanical properties, and, most important, biocompatibility, since the donor is a nitrosamine [101]. Some studies on nitrosamine-containing hydrogels are focused on the properties of the donor molecule and its potential use for antimicrobial activity, lacking further studies regarding cytotoxicity.

Biocompatible hydrogels are believed to make the overall system appropriate for topical delivery [111]. Since adverse responses are undesirable for wound healing, biocompatibility is a must for any material meant for this specific application. The repercussions of NO use and overuse on humans are yet to be uncovered. Most studies rely on in vitro tests, and the few in vivo tests are performed on mice. Mice wounds heal differently from human wounds, instead of re-epithelization, healing is made by wound edge closure. However, wound healing in pigs is made by re-epithelization and displays similar responses to growth factors. The different mechanisms might produce incongruent results when translating in vivo studies in rats to humans. Literature has shown that studies in pigs had a 78% concordance with human studies, higher than the 57% and 53% of in vitro and in vivo studies in rats, respectively [51]. Even if cytotoxic effects against mammalian cells are investigated in vitro and in vivo, complete studies of NO toxicity are needed [118].

Few systems exhibited the profile deemed best suited for wound healing, namely an initial uncontrolled release followed by a slow and constant release [68,93,112,115]. Sustained NO delivery has been reported using an antioxidant in parallel with a NONOate-based NO donor. Aiming to lessen the indiscriminate destructive power of peroxynitrite—a product of the reaction between NO and superoxide—an antioxidant was included in the formulation. The system exhibited a sustained NO release that lasted at least 12 h when complemented with the antioxidant curcumin. However, the formulation reduced collagen deposition, a process that occurs at the latter steps of wound closure [119]. Since peroxynitrite is responsible for the antimicrobial activity of NO, this system is not the most appropriate for wound healing materials. Basically, the material should remain compatible with its intended application whenever a component is added to counteract the perceived shortcomings of another component. Systems based on the addition of reducing agents (e.g., glucose, ascorbic acid) have also been reported. Ascorbic acid catalyzed a sustained NO release for the tested 36 h when used to complement a keratin-based RSNO electrospun with poly(urethane) and gelatin [63]. The higher release rate in the presence of ascorbic acid has been assigned to the reduction of Cu^2+^ to Cu^+^, as the latter enhances NO release from RSNOs [73]. The role of copper ions in solution has been postulated to increase NO release by disrupting *N*-diazeniumdiolate and amine H-bonding [27].

Kinetic studies showed that water absorption by the hydrogel controlled NO release from GSNO-loaded Pluronic F-127 and PAA hydrogels [103]. The same NO donor, incorporated in Pluronic F-127/Chitosan hydrogels and Pluronic F-127-embedded chitosan nanoparticles, followed Higuchi with Fickian diffusion kinetics [34,85], and Korsmeyer-Peppas with Fickian diffusion kinetics when incorporated in chitosan hydrogels [46]. For the NO donor *S*-nitroso-mercaptosuccinic acid in alginate hydrogels, NO release best fitted the Higuchi model with Fickian diffusion [112]. Drug release in function of time can be predicted by mathematical models. The release rate of drugs from matrix systems is described by both Higuchi and Korsmeyer-Peppas equations, the latter being a semi-empirical model used for polymeric matrices such as hydrogels. Fickian diffusion means that the drug, in this case, NO, is released by diffusion instead of swelling or polymer relaxation [120].

Although some hydrogel formulations of physically absorbed donors have been reported to exhibit no leaching [117], the possibility of leaching is higher in the case of a donor incorporated by physical adsorption. The premature and unspecific release of the NO donor from the hydrogel matrix is extremely undesired and can be avoided by coating (e.g., poly(urethane) shells) or chemical attachment of NO donors to hydrogels [85].

### 4.2. Chemically Attached NO Donors

Chemically modified NO-releasing hydrogels are based on a wide array of polymers, such as chitosan, peptides, Pluronic F-127, PVA, and gelatin (see Table 4), with chitosan as the most studied polymer to date. This polymer, as the result of deacetylated chitin, a marine polysaccharide, has a heterogeneous chemical structure composed of *N*-acetyl-glucosamine and *N*-glucosamine units. Only second to chitosan hydrogels, peptide hydrogels are composed of macromolecules that confer biocompatibility and degradability which are great for biomedical applications such as wound healing [121]. Although less common, fibrin and fibrinogen-based hydrogels have been linked to augmented wound healing. Fibrinogen hydrogels facilitate cell adhesion, angiogenesis, and cell proliferation [43], while fibrin hydrogels allow cell proliferation and can be degraded by cells intervenient in wound healing to remodel the ECM [92].

Regarding NO donors, the diversity is limited, as it consists primarily of NONOates and RSNOs, apart from a metal-NO complex [122]. *S*-nitrosothiols and *N*-diazeniumdiolates are great NO donors for these specific systems, as functional groups such as thiols and amines can react to store NO, forming RSNOs and NONOates, respectively (Figure 4). Contrary to hydrogels with physically adsorbed NO donors, leaching is absent in hydrogels with chemically attached NO donors.

**Table 4 pharmaceutics-14-01377-t004:** NO-releasing hydrogels based on covalently bound NO donors.

Hydrogel	NO Donor	NO Release Features	Reference
Poly(vinyl alcohol)	*N*-Diazeniumdiolate	~48 h	[123]
Poly(vinyl alcohol)	RSNO	Photochemical release	[39]
Pluronic F-127	RSNO		[124]
Pluronic F-127 and branched PEI	*N*-Diazeniumdiolate	Burst release in first hours, sustained up to 50 h	[77,125]
Chitosan	*N*-Diazeniumdiolate	Enzymatic deprotection by glycosidase	[126,127]
NapFFGEE peptide	*N*-Diazeniumdiolate	Enzymatic deprotection by glutathione/ glutathione *S*-transferase	[96]
Naphthalene-terminated FFGGG peptide	*N*-Diazeniumdiolate	Enzymatic deprotection	[94]
Fmoc-Pexiganan and Pexiganan	*N*-Diazeniumdiolate	~400 h	[128]
Gelatin	SNAP	Burst release in first 2 h, sustained up to 72 h	[95]
Chitosan and hyaluronic acid	SNAC	Burst release in first 2 h, sustained up to 48 h	[41]
Chitosan	*N*-Diazeniumdiolate	~48 h Enzymatic deprotection	[65]
Fibrin	SNAP	Light exposure	[92]
Laponite-poly(pentaethylenehexamine) composite	*N*-diazeniumdiolate	Burst release	[129]
Alginate modified with DETA	*N*-Diazeniumdiolate	~4 days	[80]
PEG	*S*-nitrocysteine	~24 h	[130]
Poly(caprolactone)/Poly(sulfhydrylated polyester)	RSNO		[64]
Nap-FFKEGG	*N*-Diazeniumdiolate	No burst release	[131]
Alginate and branched PEI	*N*-Diazeniumdiolate	Addition of Cu (II) increases NO release rate	[27]
Chitosan, PEG, and glucose	Nitrite SNAC		[132]
Poly(ε-lysine)	*N*-Diazeniumdiolate	~15 h	[76]
Poly(2-hydroxyethyl methacrylate)	Ruthenium nitrosyl	Photochemical release	[122]

Swift and spontaneous NO donation is observed in NONOate-modified Laponite-poly(amine) composite hydrogels, though NO release is dependent on the laponite-to-polymer ratio, further attributed to tri-dimensional disposition. NO donors intercalated between Laponite disks are less susceptible to decomposition, as interaction with the medium is delayed, therefore extending NO release [129]. A mixed release profile is reported for few systems [41,77,95,125]. Controlled and paced NO release has been reported for the systems reliant on enzymatic catalysis [65,96,126,127]. Enzymatic sensitivity allows greater control over NO delivery and release rate, as the latter is dictated by enzyme kinetics. Briefly, a NO donor is either connected to a polymer backbone and a specific enzymatic substrate, or simply to a polymeric chain. NO release occurs after the protecting molecule (e.g., galactose) is modified by the enzyme (i.e., glycosidase enzyme for galactose substrate), deprotecting the NO donor, which then decomposes. Complex and specific molecules can also donate NO upon direct modification by enzymes [96]. Moreover, enzyme-responsive delivery systems hold their cargo in the absence of enzyme independently of the medium, therefore eliminating spontaneous release.

A unique approach in which a metal-NO complex is covalently bound to the polymeric chain through polymerization in the presence of 4-vinylpyridine attached do the metal-NO complex has been reported. The resulting polymer forms a highly stable hydrogel that promptly releases NO upon UV irradiation (Figure 5). Results indicate that the photoproduct remains retained within the hydrogel structure following irradiation and NO release, further supporting the suitability of the system for biomedical applications [122].

A dual system based on a SNO-modified PVA film coupled with a GSNO-containing Pluronic F-127 hydrogel was tested for wound healing in rats (Figure 6). The film alone displayed spontaneous and fast NO release for the first couple hours, whereas the hydrogel itself showed sustained NO release. This dual phase dressing was designed to modulate NO delivery on the hypothesis that NO released from both PVA film and GSNO dispersed in the hydrogel would accumulate on the poly(propylene oxide) core of the micellar Pluronic F-127 hydrogel and be slowly released. In other words, even if NO is spontaneously released by PVA films, it will be retained in the hydrogel and slowly diffuse to the wound. Results showed that NO release occurs steadily for a period of at least 24 h when both layers are used. Moreover, the study demonstrated enhanced wound closure and a shortened inflammation phase [133].

Nitric oxide delivery would benefit from standardized results. NO is detected and measured through a variety of techniques (i.e., Griess assay, fluorescence, chemiluminescence, and electrochemical methods [50]), and independently of the technique used to detect and quantify NO release, the release profile can be easily observed with cumulative release plots. However, results remain difficult to compare because some authors report the percentage or molar concentration of released NO while others report molar concentrations of NO per mass or per area of the hydrogel. Although the total amount of NO present and released in the tested sample should always be presented, calculating partial release until NO release halts guarantees that the method and units used in NO quantification are normalized. Studies performed so far have shown that NO can indeed be stored in hydrogel matrices until release is desired and/or triggered. Since the performance of the system as a wound dressing capable of releasing NO depends on several factors (e.g., hydrogel composition, NO donor class, NO release mechanism, etc.), there is room for further studies since there are numerous formulations to choose from. Table 5 summarizes the advantages and disadvantages of the incorporation of NO donors in hydrogels and the type of mechanism behind NO release.

### 4.3. Antibacterial Activity and Wound Healing of NO-Releasing Hydrogels

Nitric oxide is an efficient antibacterial agent, and NO-releasing hydrogels have shown bactericidal efficiency against Gram-positive *Staphilococcus aureus, Staphilococcus epidermis,* and *Streptococcus mutans*, and against Gram-negative *Escherichia coli* and *Pseudomonas aeruginosa* (Table 6). Bacteria are characterized as Gram-positive bacteria when they have a thick peptidoglycan layer, and Gram-negative when their thin peptidoglycan is followed by an outer membrane that unfortunately better protects the bacteria [58,134]. The antibacterial activity of NO against planktonic and biofilm-embedded *P. aeruginosa* was tested with the use of a chemically modified antimicrobial polymer (ethylene glycol, ethylhexyl, and cationic primary amine units). In addition to great bactericidal efficacy, the NO-releasing polymer induced biofilm dispersal [87].

As for wound healing, in vivo tests performed on rats showed that NO-releasing hydrogels enhanced wound closure. A combined effect of chitosan and NO was observed as chitosan hydrogels alone accelerated wound closure and GSNO-loaded chitosan hydrogels showed superior wound healing in rats [46]. Promising results were also obtained with alginate-based hydrogels loaded with GSNO, namely accelerated wound healing and bactericidal activity against drug-resistant Gram-positive and Gram-negative bacteria such as MRSA and MRPA [40,111]. These results support the suitability of NO for wound healing purposes.

## 5. Conclusions

The interest in functional materials has been rising in recent decades. Even though quality of life has improved enormously due to the advancements in healthcare and pharmaceutics, there is room for improved materials capable of promoting wound healing. NO has garnered increased interest over the past few years due to its antimicrobial properties and has proven to be efficient on its own, by enhancing the antibacterial efficiency of antibiotics and by decreasing resistance developed by bacteria strains, which is very important nowadays since multidrug resistant bacteria strains are difficult to eradicate. NO further aids wound healing by fastening inflammation, inducing angiogenesis and facilitating cell proliferation, making it an excellent bioactive compound for wound closure. Since NO is a gas, its delivery is made through molecules or systems capable of donating NO. Multiple classes of NO donors have been explored, with *N*-diazeniumdiolates and *S*-nitrosothiols being the most prevalent due to their higher stability relative to other classes of compounds capable of donating NO. Nonetheless, vesicles or matrices are generally used to extend NO donor stability and consequent shelf-life. Besides being popular polymeric matrices suitable for NO donor storage and protection, hydrogels are also a promising class of material for wound healing purposes since these structures can physically protect the wound bed from external factors.

Two approaches are observed in the preparation of hydrogels for NO delivery: NO donors incorporated in or chemically attached to hydrogel platforms. Hydrogels with chemically bound NO donors are devoid of NO donor leaching, contrary to some hydrogels with physically adsorbed NO donors. The profile of NO delivery is influenced by the class of donor, mechanism of decomposition into NO, and hydrogel loading method. The release profile itself can be modulated to a certain point, from abrupt to continuous, or even mixed. The latter is deemed more beneficial for wound healing. Photochemical or enzymatic responsiveness provides NO deliveries with ultimate control where the release is initiated upon stimulation instead of instantaneously.

The antibacterial activity of hydrogel-delivered NO has been tested against Gram-positive and Gram-negative bacteria, normal and drug-resistant strains, with promising results. In the future, antibacterial activity should include biofilm dispersal and eradication besides the typical planktonic bacteria eradication. The performance of NO-loaded hydrogels in wound healing tested in vivo is encouraging, with fastened wound healing. Although the cytotoxicity of NO against mammal cells is studied and disclosed on almost every report, further studies should be performed to determine the long-term effects of NO exposure in humans. In sum, NO-releasing hydrogels are proven to be excellent materials for wound healing purposes. According to the results reported, the interest in NO donors is far from ending since their role in wound dressings goes beyond antibacterial activity.

## 6. Future Perspective

The studies carried out to date prove the potential of both NO as a therapeutic agent for chronic wounds and hydrogels as protectors and vectors for the delivery of bioactive agents. Although the characteristics of each type of wound make it difficult or even impossible to formulate a one-size-fits-all wound dressing, it is viable to develop an optimized system for each type of wound only by modifying the properties of the hydrogel itself (adhesion, exudate absorption, water vapor, and gas permeability). An extensive characterization of the physicochemical properties of wound dressings, NO’s release profile over time, as well as the impact of NO’s toxicity given prolonged and/or recurrent exposure is essential. It is also necessary to explore the toxicity of the system and by-products after NO donation.

## Figures and Tables

**Figure 1 pharmaceutics-14-01377-f001:**
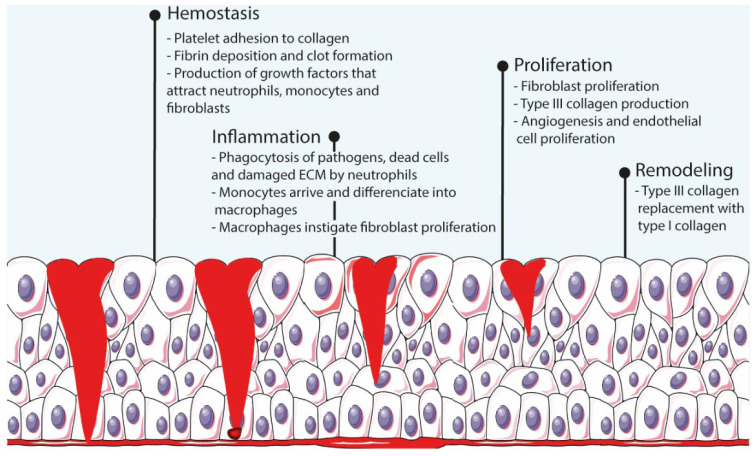
Schematic representation of the wound healing process.

**Figure 2 pharmaceutics-14-01377-f002:**
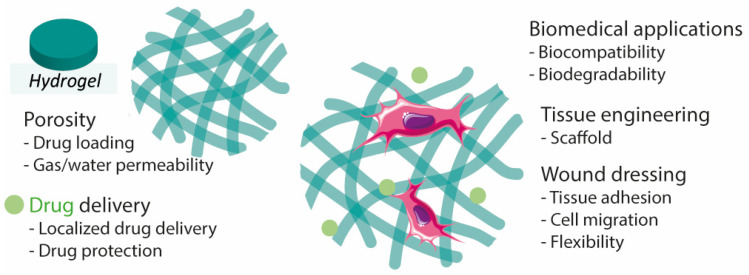
Hydrogel characteristics for wound healing, drug delivery, and tissue engineering.

**Figure 3 pharmaceutics-14-01377-f003:**
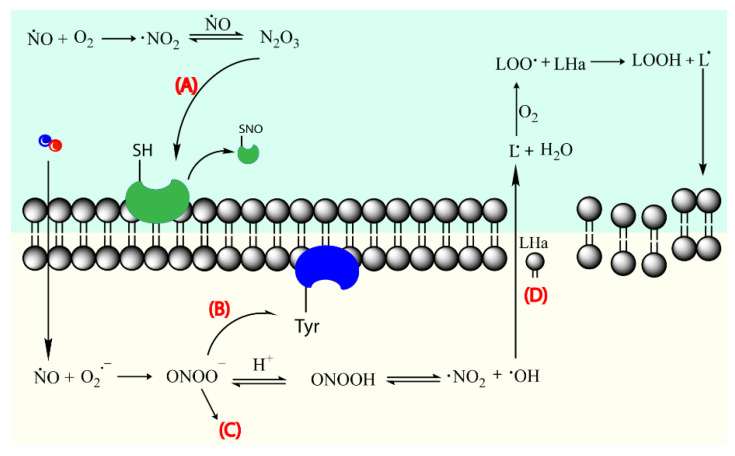
NO acting mechanism on bacteria cell membrane. Nitric oxide leads to thiol nitrosation (**A**), tyrosine nitration (**B**), DNA cleavage (**C**), and lipid peroxidation (**D**). Lipid (L) and allylic proton (Ha). Adapted with permission from Ref. [27]. Copyright 2021 American Chemical Society and Adapted with permission from ref. [54]. Copyright 2008 American Chemical Society.

**Figure 4 pharmaceutics-14-01377-f004:**
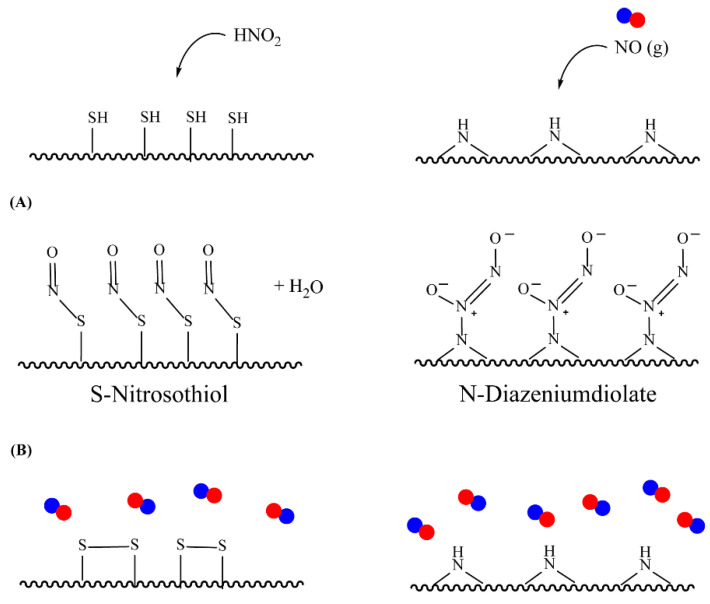
Nitric oxide capture (**A**) and release (**B**) by thiols (left) and amines (right) to originate *S*-nitrosothiols and *N*-diazeniumdiolates, respectively. Adapted from ref. [39].

**Figure 5 pharmaceutics-14-01377-f005:**
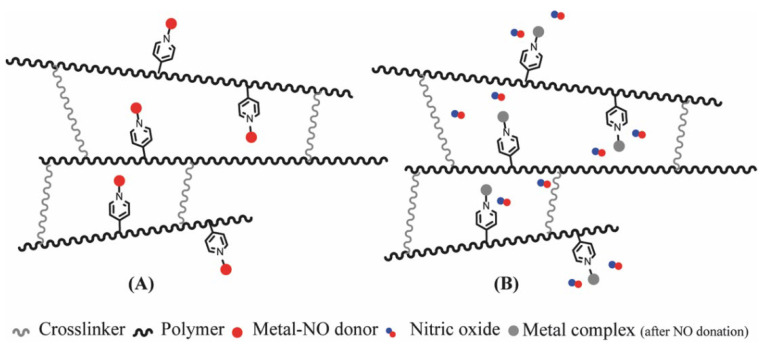
NO-releasing hydrogel with NO donor covalently linked to a pyridine derivate inserted in the polymeric chain. Before (**A**) and after (**B**) irradiation. Adapted from ref. [122].

**Figure 6 pharmaceutics-14-01377-f006:**
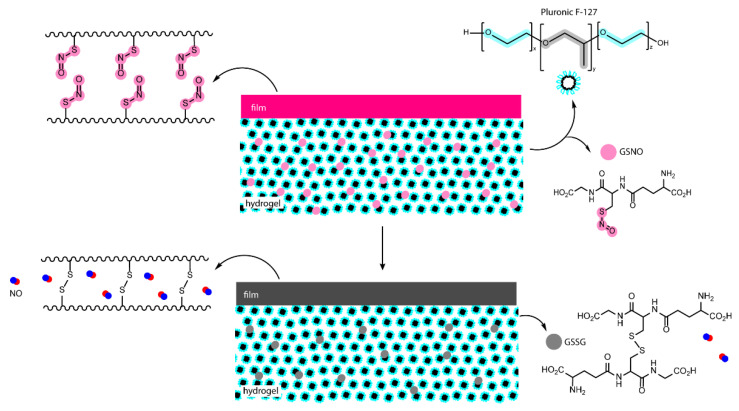
Poly(vinyl alcohol) films modified with SNO groups and GSNO-containing Pluronic F-127 hydrogels. Adapted from ref. [133].

**Table 1 pharmaceutics-14-01377-t001:** Chemical structure of the most used S-nitrosothiols.

*S*-Nitrosothiol	Chemical Structure
GSNO *S*-nitrosogluthathione	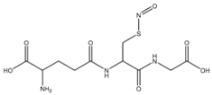
SNAC *S*-nitroso-*N*-acetylcysteine	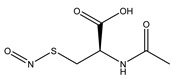
SNAP *S*-nitroso-*N*-acetylpenicillamine	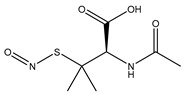
SNMSA *S*-nitroso-mercaptosuccinic acid	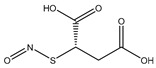

**Table 2 pharmaceutics-14-01377-t002:** Chemical structure and approximate half-life values, t_1/2_, of NONOates at 37 °C and pH 7.4. Adapted from [81].

N-Diazeniumdiolate	Chemical Structure	*t* _1/2_
PROLI/NO1-[-2-(-carboxylate)pyrrolidine-1-yl] NONOate	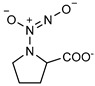	2 s
MAHMA/NOMethylamine hexamethylenemethylamine NONOate	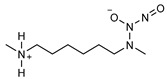	1 min
DEA/NODiethylamine NONOate	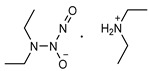	2 min
SPER/NOSpermine NONOate	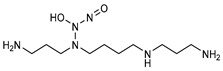	6 min
PAPA/NOPropylamine propylamine NONOate	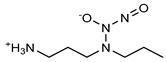	15 min
DPTA/NODipropylentriamine NONOate	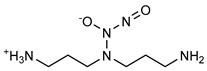	3 h

**Table 3 pharmaceutics-14-01377-t003:** NO-releasing hydrogels based on physically adsorbed NO donors.

Hydrogel	NO Donor	NO Release Features	References
pHEMA	Manganese nitrosyl	Light-activated	[82]
Methacrylate-modified gelatin /hyaluronic acid graft dopamine	*N*,*N*′-di–sec–butyl–*N*,*N*′-dinitroso-1,4-phenylenediamine (BNN6)		[101]
Gelatin methacrylate and oxide dextran	BNN6	Near-infrared release	[42]
Gelatin	Sodium nitrite		[83]
Gelatin methacrylate	SNAP (S-nitroso-N-acetylpenicillamine)		[102]
Gelatin and sodium alginate	SNAP	Burst release in first 4 h, sustained up to 120 h	[68]
F-127/PAA	GSNO	~200 min constant ~5 days	[103]
Pluronic F-127	GSNO	--------	[34,104,105,106,107]
Pluronic F-127	GSNO SNAC (*S*-nitroso-*N*-acetylcysteine)	Thermal or photochemical release	[84,108,109]
Pluronic F-127	GSNO		[85]
Pluronic F-127 and alginate	GSNO		[110]
Pluronic F-127 Pluronic P-123	Nitroso-derivative of 4-amino-7-nitrobenzofurazan	Photochemical release	[111]
Alginate, pectin and PEG	GSNO	Release for at least 18 h of GSNO	[40]
Alginate	S-nitroso-mercaptosuccinic acid	Burst release in first 5 h, sustained in following hours (tested up to 18 h)	[112]
Chitosan	Isosorbide mononitrate (ISMN)		[113]
Chitosan	GSNO	Sustained for over 48 h	[46]
Chitosan, PEG, sugar	Sodium nitrite	Sustained for at least 24 h	[44]
Chitosan, PVP, PEG	Nitrite	Burst release for 120 min followed by sustained up to 8 h	[93]
Chitosan, PVA	SNAP	Continuous release for at least 120 h	[114]
Chitosan and Poly(vinyl alcohol)	Ruthenium nitrosyl	NIR-induced release	[90]
PEG, fibrinogen	SNAP	Photolytic and thermal activation	[43]
Fmoc-FF	SNAP	Burst release in the first 12 h, sustained over 7 days	[115]
Poly(β-cyclodextrin) and modified dextran	Nitro compound	Photochemical	[91,97,116,117]

**Table 5 pharmaceutics-14-01377-t005:** Summary of advantages and disadvantages of NO donor incorporation in hydrogels and NO release mechanisms.

	Advantages	Limitations
Mechanism	NO Donor Incorporation	
Physical adsorption	Simple, no reactions or modifications required Any NO donor can be incorporated	Possible leaching Storage, stability, and release depend on hydrogel–donor interactions
Chemical attachment	No leaching Ease to create RSNOs and NONOates	Requires complex reactions
	NO release	
Hydrolysis	Uncomplicated release triggers	Undesired release in water containing environments
Enzymatic catalysis	Not subject to uncontrolled release due to specific triggers	Release rate depends on enzyme kinetics
Photocatalysis		Limited application, requires direct irradiation

**Table 6 pharmaceutics-14-01377-t006:** Antibacterial activity of NO-releasing hydrogels against Gram-positive and Gram-negative bacteria, with a focus on the systems with enhanced wound healing tested in vivo. Methicillin-resistant *S. aureus* (MRSA) and multidrug-resistant *P. aeruginosa* (MRPA).

Gram +	Gram −	Effect	NO Donor/Hydrogel	References
Antibacterial activity assessed in vitro				
*S. aureus*	*E. coli*	Bactericidal	Metal-NO complex/Chitosan, PVA	[90]
	*E. coli*	Bactericidal	Nitro compound/Poly(cyclodextrin)	[91]
*S. epidermis*	*E. coli*	Bactericidal	NONOate/Chitosan, Hyaluronic acid	[41]
*S. mutans* *S. aureus*	*E. coli*	Bactericidal	RSNO/Alginate	[112]
	*P. aeruginosa*	Bactericidal	GSNO/Chitosan, Pluronic F-127	[34]
*S. aureus*	*P. aeruginosa*	Bactericidal	NONOate/Alginate, PEI	[27]
	*P. aeruginosa*	Bactericidal Biofilm dispersal	NONOate/antimicrobial polymer	[87]
With enhanced wound healing tested in vivo				
*S. aureus*	*E. coli*	Bactericidal	BNN6/GelMA	[42]
*S. aureus*	*P. aeruginosa*	Bactericidal	GSNO/Chitosan	[46]
MRSA	*P. aeruginosa*	Bactericidal	GSNO/Alginate, Pectin, PEG	[40]
MRSA	MRPA	Bactericidal	GSNO/Alginate	[110]

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
