# Peer review of "Recent Advances in Hydrogel-Mediated Nitric Oxide Delivery Systems Targeted for Wound Healing Applications"

_pharmaceutics, 2022, doi:10.3390/pharmaceutics14071377_

Round 1

Reviewer 1 Report

Following the revision the overall manuscript quality has improved.

However, some issues still remain, the vast majority of the manuscript focuses on NO donors and anti-bacterial actions. Now the authors have chosen to change the title of their manuscript to include 'targeted wound healing applications' 

The effects on actual endogenous cell wound healing are hidden in a small section (lines 229-239). Instead, the authors should reinforce the anti-bacterial role in their title. e.g. 'targeted healing in bacteria infected wounds' or similar.

When referring to wound healing actions (i.e. endothelial cell stimulation, stem cell regulation, immune cell regulation etc.), since the authors don't write about these actions they should refer readers to in depth reviews on these actions whilst stating that those roles are outside of the current topical review's scope.

These minor adjustments will help to sharpen up the manuscript's unifying message and sub-section connection/flow from start to finish.

Reviewer 2 Report

In this manuscript, the authors reviewed the "Recent advances in hydrogel-mediated nitric oxide delivery systems targeted for wound healing applications". In my opinion, some issues should be further addressed and I hope the following comments could be helpful for improving their paper.
  1.  The introduction section needs to be rewritten, no need for a sub-heading in the introduction section. The authors should enrich this part and emphasize the necessity of "nitric oxide delivery via hydrogel" for wound healing applications
  2. Authors focused on wound healing applications, but what are the distinguished properties and specific problems of wound healing applications? The authors never discussed it.
  3. According to the applications of hydrogel-mediated nitric oxide delivery systems, most, if not all are applicable for other kinds of diseases. Then why did the authors not expand the topic to other diseases?
  4. Good quality figures are very important for a  review paper, kindly improve the quality of the figures and try to add at least 4-5 figures in this manuscript from the recent literature and the authors need to point out the main idea for each figure in the text.
  5. The authors should summarize the current approaches to fabricating "hydrogel-mediated nitric oxide delivery systems" and compare their advantages and disadvantages in order to draw the reader's attention.
  6. This manuscript is well organized but lacks specific comparative analysis. What are the advantages of "hydrogel-mediated nitric oxide delivery systems" compared with traditional technology?
  7. Please revisit the entire manuscript for minor grammar issues.
  8. Future perspective is very important for a review paper, kindly add this heading in your review paper briefly, the author should consider giving some methodological design about how to improve the performance of hydrogel-mediated nitric oxide delivery systems
  9. A more rigorous analysis of unintended consequences and the risk/benefit balance is needed. For example, nitric oxide is generally very dangerous and damaged normal cells. Testing the biocompatibility of hydrogel-mediated nitric oxide delivery systems deserves more attention. 
    4. Discussing drug delivery by cells, one should note the lack of rigorous studies of their biodistribution, trafficking, PK and longevity of cell-loaded cargoes. The only exception is RBC-drug delivery, and even in this relatively established area, most studies omit the quantitative analysis of PK and BD of the carrier and cargo.
    10. Please revisit the entire manuscript for minor grammar issues.

Round 2

Reviewer 2 Report

Accepted in present form 

This manuscript is a resubmission of an earlier submission. The following is a list of the peer review reports and author responses from that submission.

Round 1

Reviewer 1 Report

Dear authors, 

In the following, please find my comments and suggestions.

Section 1:

The statistical data published by the international healthcare organizations regarding the healthcare-associated infections and deaths are outdated (References 7 and 8 contain 5-10 years old data). The authors must search for updated statistics.

The organization of the text is a little bit confusing. There is Section 2 with Subsection 2.1 and that’s all.

The same structure has also the Section 3.

I recommend to the authors to reorganize these sections.

Section 2:

The antimicrobial effect of NO is sustained by few (8) references and I consider that for a “review” type this number is small and must be increased with some more recent data. Also, the authors have to consider that is in the manuscript interest to be improved with other relevant examples of NO’s effect against different microorganisms.

The efficiency of NO against biofilm-inserted or planktonic bacteria must be proven by valuable updated references, which in this form are missing. On the other hand, the antibacterial mechanism is not sufficient described.

The paragraph devoted to S-nitrosothiols (from 124 to 141 lines) is sustained by a small number of references (only 4).

As a remark, reference no: 20 is cited many times (8), but not in a clear mode. This is a book with 13 chapters having as common subject “Nitric oxide” and is very difficult to find the information that our authors are referring to. My suggestion is to correlate the data that appear in the manuscript with the concrete reference (chapter) in order to follow-up easy.

Section 3:

In my opinion, the authors must improve the paragraphs referring to the “two functions” of the hydrogels (lines 193-194) as well as the ways how to implement hydrogels (lines 196-197). The references (especially, updated!) that could support these characteristics of hydrogels are completely missing.

Reviewer 2 Report

The review by Tavares, Alves, and Simoes provide a good amount of detail and an overview on NO-donors and NO-donor functionalised hydrogels. Despite a in-depth coverage of NO-releasing materials, the review lacks scope and structure, which could be remedied by re-writing the introduction sections.

The problem lies in the authors choice to limit themselves and focus on dermal wound healing (mainly antibacterial function) of NO in the first two introduction paragraphs. These are loosely connected to the rest of the manuscript. There is no mention of wound healing function in the review title, and a lot of the hydrogel materials discussed were utilized for the repair/regeneration of multiple tissue types, demonstrating a diversity of functions - most notably of course, angiogenesis/vascularisation, which has imperative functions in wound healing.  

It is suggested that the authors further expand the introduction section (and abstract). The dermal wound healing applications, and antibacterial activity are just two sub-sections of the diverse roles that NO plays in mediating tissue repairs processes in multiple tissues. The review would benefit from a more general overview of the multiple and emergent functions of NO (and summary figure) that make it such a desirable molecule to exploit in its controlled release from hydrogels for enhanced tissue repair.

The NO therapy for tissue repair/regen field is still booming and achieving high impact works, some suggestions include:

A brief overview of NO's emergent functions in regulating stem cell function and mediating inflammation [example dois: Acta Biomater 10.1016/j.actbio.2017.08.037; Sci Rep 10.1038/srep08718; Stem Cell Dev 10.1089/scd.2013.0646; etc.].

A brief overview of NO's critical roles in driving neo-vascularisation, repair of ischemic tissues, and potential in chronic diseases [example dois: Frontiers Bioeng 10.3389/fbioe.2021.770121; Adv Healthcare Mater 10.1002/adhm.201801210; etc.].

and as an attachment related to wound healing, brief coverage of proposed anti-cancer effects [example dois: Nature Nanotech 10.1038/s41565-019-0570-3; Vaccines 10.3390/vaccines9020094; etc.].

These sections would also serve to introduce what is widely considered to be key of NO therapy - concentration/dose and situational release.

The conclusion section would be substantially strengthened if the authors discussed advanced/cutting-edge materials or promising technologies and their future perspectives for achieving controlled release of suitable conc. of NO, which is one of the key bottlenecks facing NO-releasing material development. What core properties makes hydrogels desirable for NO-release? What would an idealistic hydrogel require? 

Reviewer 3 Report

The paper entitled “Recent advances in hydrogel-mediated nitric oxide delivery 2 systems” by Tavares et al. (pharmaceutics-1679894), is interesting, but I have same suggestions for the authors:

The introductive section (Section 1) should contain more relevant and complete information on wound care management. See below some aspects which must be improved:

Even if microbial contamination represents a major factor for delayed wound healing, the authors should include other relevant examples that contribute to this unwanted effect (e.g.: inflammation, oxygen deficiency, immune-/metabolic-/vascular-related pathologies…)

Lines 19-20: The phrase sounds rather redundant. A more concise and complete definition of wounds should be provided.

Line 22: “Acute wounds tend to heal relatively fast, while chronic wounds require…” – the authors could mention some etiological factors that determine the occurrence of acute and chronic wounds.

Lines 31-34: The authors should provide more recent statistical data, according to international healthcare organizations. Refs 7 and 8 contains data during 2011-2012 and 2016-2017 time periods, respectively.

Lines 35-36: In my opinion, this information is incomplete, as the “proper wound care” considers more than wound dressings (e.g.: proper debridement, wound perfusion / nutrition, oxygen therapy…).

Lines 48-52: The authors should include some exact requirements of the “proper wound dressing”, by addressing the porosity, gas / water permeability, detailed biocompatibility-related characteristics.

Lines 57-63: More recent and relevant references should support the information within. Also, the advantages and limitations of different antimicrobial agent – modified “porous materials” should be mentioned. The authors are encouraged to point out the role of “other components that accelerate the healing process”, as they in-text mentioned biomolecules act through different mechanisms.

Lines 64-69: Before mentioning the use of biopolymers in wound dressings, I highly recommend the authors to mention – at least – why they are preferred over synthetic polymers, and even to include some pros and cons for both natural and synthetic polymers.    

Section 2:

Lines 71-73: I recommend the authors to include some recent data on the antimicrobial effects of NO (with relevant examples against different microorganisms), from the past 5 years.  

Lines 78-80: This assertion should be supported by more recent references. Moreover, fibroblasts are not the only type of cells that play an active key role during wound healing process; therefore, a highly recommend the author to include similar data on other cells (e.g.: macrophages, keratinocytes, endothelial cells…).

Lines 88-96: In my opinion, the information included within this paragraph must be supported by recent and multiple references, which evidence the efficiency of NO against both planktonic and sessile bacteria. In addition, the antibacterial/antimicrobial mechanism(s) should be discussed.

There is subsection 2.1. But only 2.1, with no other 2.x subsections…

Lines 124-141, 144-162, 167-172: In my opinion, the information included within these paragraphs should be supported by multiple references, but also be more precise in the case of lines 167-172.

Section 3:

Lines 190-193: The information should be expanded, as to include other relevant examples of “improved or enhanced properties”.

There is subsection 3.1. But only 3.1, with no other 3.x subsections…

Hydrogel formulations mentioned in tables 3 and 4 have some common major elements (e.g., gelatin, chitosan, PVA, PEG…). I strongly consider that it is important for the authors to mention some advantages / benefits / limitations / comparisons of these formulations, as the tabular data provide some essential (but incomplete) characteristics.

I do consider that some information on the in vitro / in vivo performance of NO-releasing hydrogels should be highlighted.